# A deep learning framework for gait-based frailty classification using inertial measurement units

Arslan Amjad[1]*, Agnieszka Szczęsna[1], Monika Błaszczyszyn[2], Jerzy Sacha[2,3], Magdalena Sacha[4], Piotr Feusette[3], Wojciech Wolański[5], Mariusz Konieczny[2], Zbigniew Borysiuk[2], Basheir khan[6]

1 Department of Computer Graphics, Vision and Digital Systems, Faculty of Automatic Control, Electronics and Computer Science, Silesian University of Technology, Gliwice, Poland, 2 Department of Physical Education and Sport, Faculty of Physical Education and Physiotherapy, Opole University of Technology, Opole, Poland, 3 Department of Cardiology, University Hospital, Institute of Medical Sciences, University of Opole, Opole, Poland, 4 Department of Family Medicine and Public Health, Institute of Medical Sciences, Faculty of Medicine, University of Opole, Opole, Poland, 5 Department of Rehabilitation, University Hospital, Institute of Medical Sciences, University of Opole, Opole, Poland, 6 Faculty of Science, Institute of Mathematical Sciences, Universiti Malaya, Kuala Lumpur, Malaysia

* aabbasi@polsl.pl

## Abstract

Frailty in older adults leads to heightened vulnerability to adverse health outcomes, significantly burdening individuals and society by increasing healthcare costs and dependency. To address this issue, an advanced frailty assessment method combining wearable sensors measurements with Deep Learning (DL) techniques is proposed to classify individuals into frail or non-frail stages. Wearable sensors provide real-time monitoring, facilitating early detection and timely interventions. Two diverse datasets, i.e., GSTRIDE and FRAILPOL, were utilized for enhanced frailty analysis, employing one to five Inertial Measurement Unit (IMU) sensors with varying configurations and mounting positions. A participant-centric data partitioning framework based on signal windows segmentation is proposed and applied to DL algorithms. Among the DL algorithms, InceptionTime outperformed, achieving 82% accuracy on GSTRIDE and 79% on the FRAILPOL dataset. Furthermore, the area under the ROC curve (AUC) and evaluation metrics such as precision, recall, and F1-score confirm InceptionTime's effectiveness in classifying frail and non-frail stages by capturing spatio-temporal features from raw IMU signals.

## 1. Introduction

Frailty, a physical syndrome, is characterized by a decrease in physical functions such as decreased muscular strength, balance, walking ability, and accidental weight loss [1]. The older adults are mostly affected by this syndrome, which can lead to serious health problems that include falls, incapacity, hospitalizations, and even death [2].

**Data availability statement:** The data supporting the findings of this study are from two sources. The GSTRIDE dataset is publicly available at https://zenodo.org/records/6883292 (García-de-Villa, S., et al. (2023)). The second dataset, FRAILPOL, is publicly available on the figshare repository (https://doi.org/10.6084/m9.figshare.c.7874411).

**Funding:** The author(s) received no specific funding for this work.

**Competing interests:** The authors have declared that no competing interests exist.

Conventional methods for assessing frailty include physical performance tests and clinical evaluations. The most widely used methods are Fried's Frailty Phenotype [3], Timed Up and Go (TUG) Test [4,5], and Short Physical Performance Battery (SPPB) [6]. The factors that are considered in the Frailty Phenotype tests are: accidental weight loss, poor physical activity, self-reported tiredness, weakness (measured by grip strength), and slow walking speed. The TUG test assesses the time it takes for a participant to stand up from a seated position, walk a short distance, turn around, walk back, and sit down again. It is sometimes combined with other variables, such as grip strength. The SPPB uses balance tests, walking speed evaluations, and repeated chair stands to evaluate lower extremity function and overall frailty.

However, the practical limitations of the traditional frailty evaluation approaches include the need for specific clinical settings, trained staff, and significant time and space resources [7,8]. In contrast, recent review studies [9,10] have shown the shift of researchers trends towards wearable sensors, particularly Inertial Measurement Units (IMUs), combined with Machine Learning (ML) or Deep Learning (DL) methods for assessing frailty. These technologies provide objectivity, precision, portability, and cost-effective techniques for monitoring physical function and gait metrics, supporting early detection and prevention of frailty in older adults [11,12].

Although frailty assessment techniques have advanced, questions remain about the optimal use of wearable sensor-based ML or DL techniques. These questions include the number of sensors required, their best mounting positioning, the parameters to be extracted, data partitioning into training and testing sets, and the algorithms for processing IMU gait data.

In this study, the research questions are addressed by a proposed DL framework that utilizes the combination of the sliding window [13] technique and a participant-based data partitioning approach. Previously, the sliding window technique was used in frailty analysis to segment the time-series data for smaller and manageable windows. However, a significant gap lies in most of the studies to address the critical issue of data leakage while using this technique. Data leakage frequently occurs if the sliding window approach is used without caution, which separates data from the same participant into the training and validation sets. This causes the model to overfit and have limited generalizability during training, as it learns patterns particular to individual participants rather than generalizable trends in the population. This issue is addressed in this study by implementing a participant-based framework that ensures no overlapping of participants' windows data in training and testing sets. This proposed framework enhances the robustness and effectiveness of DL models in frailty classification. Additionally, it reflects a real-world frailty assessment clinical condition where data from different individuals should be used independently to evaluate model performance. The proposed framework not only improves the DL model's effectiveness but also provides a more reliable foundation for clinical decision-making in frailty evaluations.

The proposed methodology is evaluated on two diverse datasets. The diversity is in terms of sensor count and the mounting position of the sensors on the participant's body limbs. The empirical evaluation of the proposed framework depicts

its effectiveness and the identification of optimal DL algorithms for diverse datasets. Other research questions that are addressed in this study are: optimal sensor usage, mounting positioning, parameter extraction, and algorithm selection for processing raw IMU gait data.

This study aims to classify signal-segment into two stages, namely frail or non-frail (robust), by implementing the following objectives:

1. Evaluate two diverse datasets, each with a different number of IMU sensors mounted on the participants' body parts. This helps to understand how diverse sensor configurations impact the frailty classification task.

2. Evaluate the most effective DL architectures for processing raw IMU gait data acquired from various sensor configurations. The goal is to increase the accuracy and precision of frailty classification.

3. Develop a participant-based framework to ensure that DL models are robust and generalizable to different participants.

By addressing these objectives, this study can contribute to improving the efficiency and reliability of frailty assessment models. In clinical settings, it also advances the use of DL and wearable sensor technologies for the early identification and prevention of frailty in older individuals.

This paper is structured as follows: In section 2, a comprehensive review of relevant work is covered; description of research methodology, datasets, and DL architectures is discussed in section 3; results are shown in section 4; and discussion is detailed in section 5. While general conclusions are drawn in the conclusion section.

## 2. Relevant work

Recent studies have highlighted the significance of using DL techniques with raw IMU sensor-based gait data. The studies generally rely on single or multiple IMU sensor data with random data partitioning techniques to analyze frailty syndrome, such as frailty analysis, fall detection, loss of balance, and gait disorder analysis.

A study [14] utilized an IMU sensor with a DL network of LSTM and CNN to classify frail and pre-frail among 20 participants. A random partition of training, validation, and testing data gave a maximum accuracy of 95% on the TUG model. Images generated from the raw IMU signals and then feed those images to DL algorithms is another effective approach for frailty analysis that was explored in studies [15–17].

Similarly, in other frailty syndrome analysis applications, wearable sensors (i.e., IMU sensors) played an important role when combined with DL approaches. A study classifies gait disfunctions among 50 participants using multiple sensor data. The data was partitioned randomly into training and testing sets with a ratio of 80:20. CNN and random forest models were applied and achieved accuracy of 88.40% and 92.30%, respectively [18]. A study [19] detected loss-of-balance in eight older adults who wore three IMU sensors while performing daily activities. With a leave-one- participant-out strategy, the BiLSTM model outperformed with an AUROC score of 0.87.

Additionally, in a fall classification task, researchers utilized multiple IMU sensor gait data with DL approaches. The study [20] used two IMU sensors mounted on each foot of participants and an LSTM model to classify fall risk, achieving a classification accuracy of 92.10%. In research [21], fall risk was evaluated in 28 participants using six IMUs sensor. Five conventional ML models and a hybrid CNN-LSTM model were utilized for this task. The CNN-LSTM model achieved a maximum F1-score of 95.18% using random data split with 5-fold cross-validation. Similarly, the study [22] utilized three wrist-worn devices with LSTM and transfer learning techniques to achieve a F1-score of 93% with random data split. Another study [23] implemented convolutional and bidirectional LSTM algorithms to learn spatio-temporal features from two smartwatches equipped with IMU sensors, achieving 88.90% accuracy in fall risk classification. In study [24], an improved and efficient CNN-BiLSTM model was assessed for fall risk using fusion of multi-sensor data (i.e., planter pressure sensor with two IMU sensors). The model achieved an accuracy of 98.40%, using plantar pressure sensor data of 24 individuals.

The studies [25–27] exploited the DL algorithms with single IMU sensor data for fall risk analysis. In a study [25], the ensemble algorithm (CNN-BiGRU) achieved an F1-score of 98% to classify falls among 32 participants. In another study [26], smartphone-based inertial gait data with Fully Convolutional Neural Networks (FCNNs) and transfer learning achieved a ROC-AUC of 93.30%. The study [27] applied data augmentation and dropout techniques to time-series IMU data and obtained 98.40% accuracy.

The IMU sensor's data utilization with DL algorithms varied in the previous studies. This literature review suggests the need for further improved methodologies, particularly in the data partitioning techniques for raw IMU gait data, to enhance the efficiency of frailty analysis in a real-world clinical environment.

## 3. Methodology

The research methodology adapted in this study is based on a DL framework that integrates a sliding window-based temporal segmentation technique with participant-centric data partitioning to ensure subject-independent model validation. Raw IMU gait data from two separate datasets was utilized to classify the signal segments as frail or non-frail (robust). The datasets differed by the sensor count and the mounting position of the sensors on different body parts of the participants. A publicly available dataset consists of a single IMU sensor, whereas the proprietary dataset utilizes five IMU sensors. Initially, the raw IMU signals extracted from both datasets were pre-processed by noise reduction, segmentation, and standardization. Then the pre-processed signals were passed through a proposed participant-based framework to divide the data into training, validation, and testing sets. These datasets were evaluated with DL algorithms to classify the signal segments as frail or non-frail (robust). Fig 1 illustrates the systematic approach of the proposed methodology from raw IMU signals to the frailty classification.

### 3.1. Datasets

**3.1.1. GSTRIDE dataset.** GSTRIDE is a publicly available dataset [28], which includes health assessments and gait data from 163 elderly participants (118 women and 45 men, ages 70–98 years, average BMI of 26.1±5.0 kg/m²). It provides comprehensive socio-demographic details (age, gender, and participants living environment), anatomical metrics (weight, height, BMI), and cognitive assessments using the Global Deterioration Scale (GDS). The dataset includes standardized tests for assessing frailty, including the 4-meter Gait Speed Test, Hand Grip Strength assessment, SPPB, TUG test, and Short Falls Efficacy Scale International (FES-I).

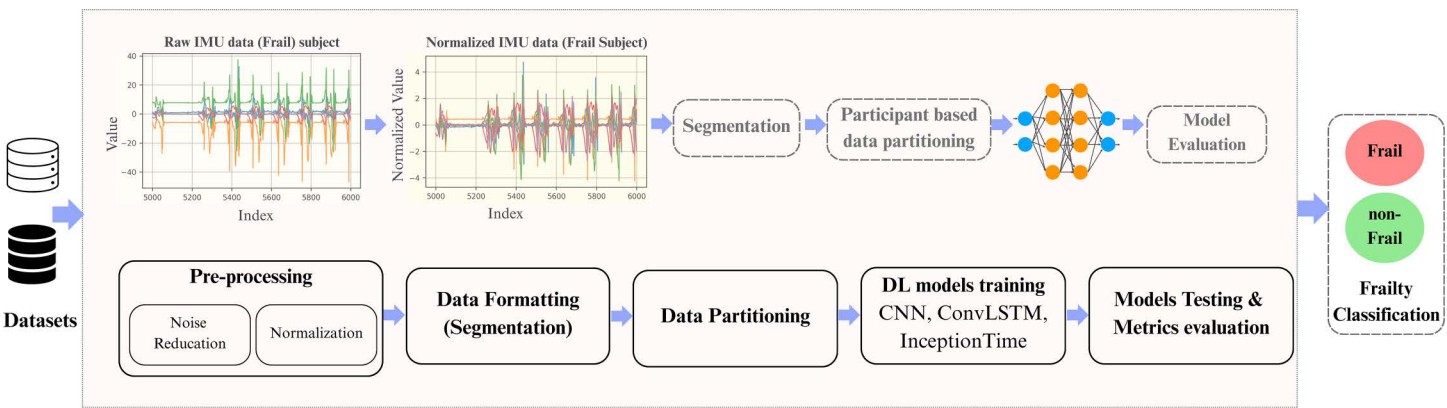

**Fig 1. A systematic methodology approach from raw IMU signals to DL models evaluation.**

In the GSTIDE dataset, a walking test was conducted with an IMU sensor attached on the foot. Two types of IMUs were used: CSIC and Gaitup, with only one sensor put on the foot at a time. The sensor captured raw IMU signals, including 3D accelerometer and gyroscope data with a frequency of 104 Hz. According to the authors [28,29] variations in sensor specifications and sampling frequencies had minimal impact on spatio-temporal estimations, ensuring the reliability of the raw IMU gait recordings for analysis in this study.

**3.1.2. FRAILPOL dataset.** The dataset of 668 participants was collected by the authors and is publicly available on the figshare repository (https://doi.org/10.6084/m9.figshare.c.7874411). The participants ages range from 65 to 89 years, with an average BMI of $28 \pm 2 \, \text{kg/m}^2$. The data gathering procedure began in 2017 and lasted five years. The FRAILPOL dataset consists of various frailty evaluation factors, such as basic medical examinations, cognitive function tests, psychosocial evaluations, dietary assessments, activities of daily living, and fall risk assessments. The National Health Foundation's electronic system was utilized to validate the outcomes. The research was registered as Clinical Trial NCT04518423 (https://ichgcp.net/clinical-trials-registry/NCT04518423).

The database contains many demographic and health features, such as cognitive health data and assessment results from Fried's phenotypic [3] and TUG [4] tests. Demographic data includes date of birth, age, and gender. Cognitive health data provides weight, BMI, MMSE scores, and pulse rate. Participants' frailty status was assessed using Fried's phenotypic test, which considers variables such as weakness, slowness, tiredness, and poor physical activity.

FRAILPOL dataset was acquired using five Xsense IMU sensors [30] during the TUG [4] test. Each participant was equipped with IMU sensors mounted on their body limbs: two wrist sensors, two ankle sensors, and one sensor at the back of the sacrum, as illustrated in Fig 2. These sensor placements were crucial for capturing the entire upper and lower body kinematics [31]. The sensors recorded 3D accelerometer and gyroscope data at a frequency of 100 Hz, including roll, pitch, and yaw information. Each participant's data was systematically organized into five text files, uniquely identified by subject ID and sensor position.

This research utilized the raw IMU signals from the 3D accelerometer and gyroscope from both datasets to analyze the frailty. This results in enhanced precision and reliability of the frailty classification model. A detailed comparison of the GSTRIDE and FRAILPOL datasets is shown in Table 1, which highlights the key parameters and frailty assessment methods of each dataset.

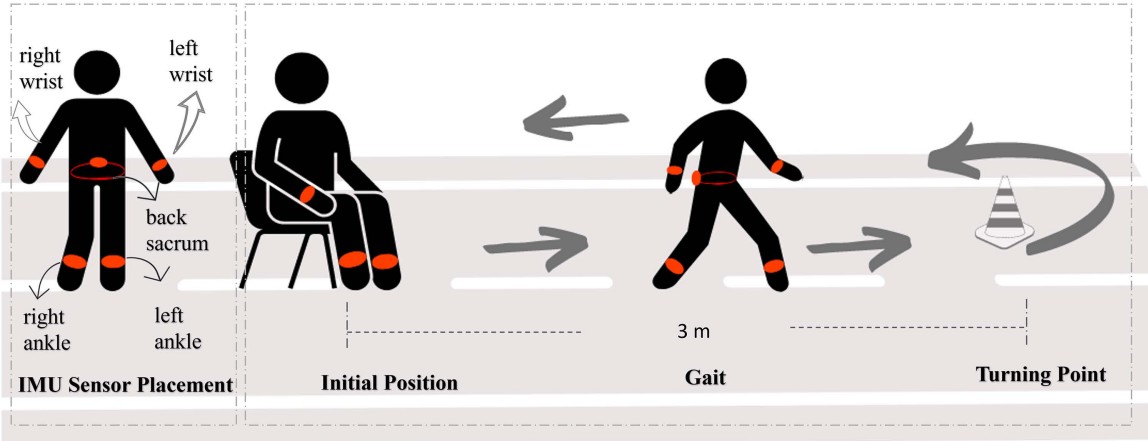

**Fig 2. Visualization of TUG test, a data acquisition in FRAILPOL using five IMU sensors mounted on participant's body limbs.**

**Table 1. Overview of GSTRIDE and FRAILPOL datasets.**

| Dataset | Partici-pants | IMU Sensors | Sensor Placement | Gait Data Collected | Other Assessments | Usage in Study |
|---------|---------------|-------------|------------------|---------------------|-------------------|----------------|
| GSTRIDE [28] | 163 | 1 | Foot | 3D accelerometer and gyroscope | Socio-demographic, anatomical metrics, GDS, Gait Speed Test, Hand Grip Strength, SPPB, TUG test & FES-I | Raw IMU gait signals for frailty classification |
| FRAILPOL (https://doi.org/10.6084/m9.figshare.c.7874411) | 668 | 5 | Wrists, ankles, sacrum | 3D accelerometer, gyroscope, roll, pitch & yaw | Cognitive function tests, Psychosocial evaluations, Fall risk assessment, Fried's phenotype test & TUG test | Raw IMU gait signals for frailty classification |

## 3.2. Frailty assessment criteria

Class labels of each participant was determine utilizing Fried frailty phenotype or Fried scale [3] method. The Fried scale is a mostly used frailty measuring tool that consists of five parameters [28,32]. Each parameter is scored either 0 or 1, and the final Frailty Index (FI) score ranges from 0 to 5, which is derived from these parameter scores. An FI score of 0 indicates that a participant is non-frail (robust), whereas an FI score between 1 and 5 suggests a frail class. The class labeling criterion is presented in Eq. (1). In our study, FI scale identified 342 non-frail and 326 frail participants in the FRAILPOL dataset. While in the GRSTRIDE dataset, 65 participants were non-frail and 98 frail.

$$ClassLabel = \begin{cases} Non-Frail, & FI = 0 \\ Frail, & 1 \leq FI \leq 5 \end{cases}$$

(1)

## 3.3. Data pre-processing

The pre-processing stage included key steps: initially, filtering out noise from the IMU signals was carried out to improve the clarity and accuracy of the data. A 4th-order zero-phase Butterworth low-pass filter with cutoffs at 15 Hz for accelerometer and 10 Hz for gyroscope was implemented using Python's SciPy's *filtfilt* function [33]. Next, class labels were assigned to each participant based on their cumulative FI score, as detailed in Eq. (1). The data was then converted into CSV format and standardized using the *StandardScaler* function for consistency.

For data formatting, the Sliding Window [13] technique was applied with a *window size* of 200 (i.e., 2 s) and a *'stride size'* of 50 (i.e., 0.5 s step). The GSTRIDE training data has a shape of (265429, 200, 6), structured as (total windows, window size, features). Similarly, for the FRAILPOL dataset the training data has a shape of (128770, 200, 7). The configuration was determined to be best following an exhaustive brute-force search that assessed a wide range of window sizes and stride rates. Both datasets include six features derived from the accelerometer ($A_X$, $A_Y$, $A_Z$) and gyroscope ($G_X$, $G_Y$, $G_Z$) sensors. The FRAILPOL dataset incorporates an additional *'Sensor Position'* feature, indicating the placement of sensors. The pre-processing pipeline was implemented in Python, using Spyder 3.10 as the development environment with built-in functions to efficiently execute these tasks.

## 3.4. Data partitioning

Traditional frailty analysis studies [34–36] with DL techniques frequently segment this time-series IMU data into windows and randomly distribute them into training, validation, and testing sets using the *'train_test_split'* function. This approach can lead to data leakage, where data from the same participant is distributed across different splits. This leakage can cause overfitting as the model may learn participant-specific patterns instead of generalizable features.

To address this issue, a participant-based data splitting framework was implemented, as shown in Fig 3. Initially, the raw IMU data extracted from both the GSTRIDE and FRAILPOL datasets was applied to the sliding window technique

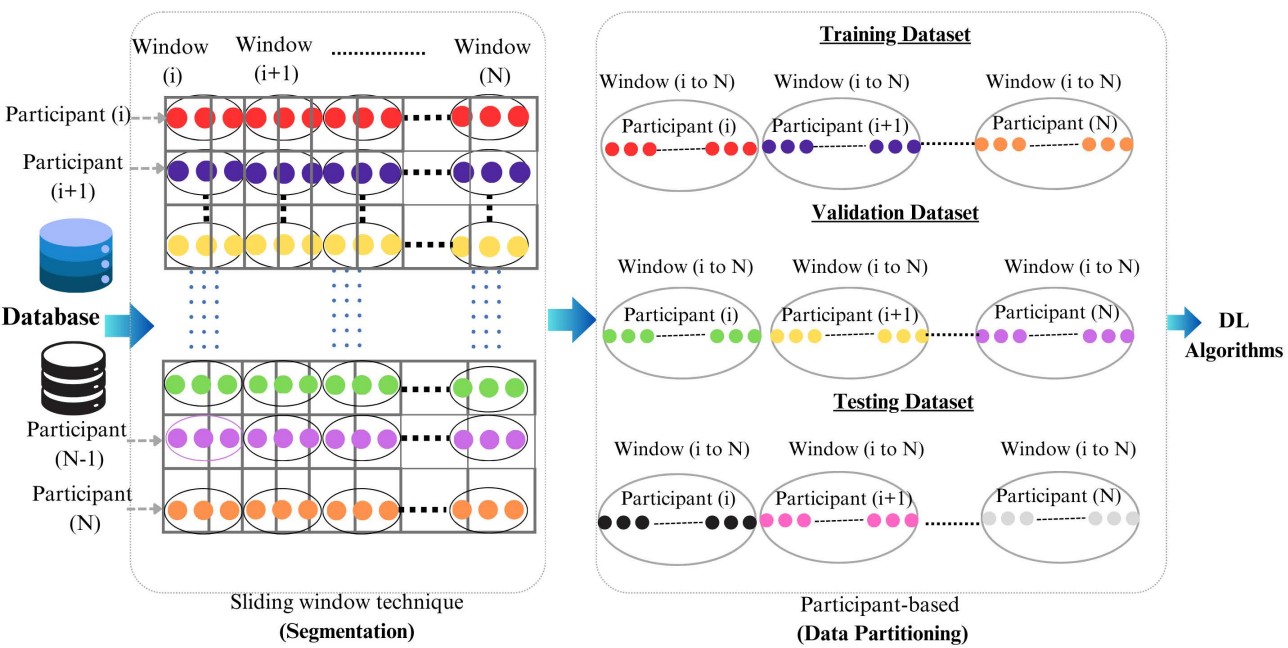

**Fig 3. IMU data partitioning for GSTRIDE and FRAILPOL datasets using the Sliding Window Technique and Participant-based splitting.**

(as mentioned in the data-preprocessing section). This data segmentation technique for the time-series data maintains the temporal information within each window. In the next step, all the windows created by the individual participants were divided into either training or validation sets with no overlapping. It is ensured by first grouping all windows belonging to the same participant and then collectively assigning the windows to either training, validation, or testing sets. No single participant's data was present in multiple partitions, ensuring total segregation between sets at the participant level. This participant-centric framework ensures that the data leakage problem during the data partitioning process is prevented, thereby improving the DL model's generalizability.

The datasets were partitioned by first creating the test set. The Stratified random sampling approach was utilized to select 10 out of 163 participants in the GSTRIDE dataset and 100 out of 682 participants in the FRAILPOL dataset, ensuring proportional class representation. The remaining participants were partitioned in an 80:20 ratio using the stratify *train_test_split* function on the array of unique participant IDs for training and validation sets, respectively. A stratified splitting method was applied to preserve consistent class distributions across all data splits.

### 3.5. Deep learning methods and implementation

Three DL algorithms, i.e., Convolutional Neural Networks (CNN) [37], DeepConvLSTM [38], and InceptionTime [39], were employed in this study to classify frailty. These models were developed using the Mcfly library [40] in Python, using Spyder version 3.10 as the development environment. The library helps to create complex DL models and optimize the hyperparameters for time-series data [41]. Separate training and validation processes were performed for both the GSTRIDE and FRAILPOL datasets using the proposed participant-based data partitioning framework. Two different configuration architectures were developed for each DL algorithm. The models were trained for 50 epochs with a batch size of 64 and the early stopping patience set to 3.

The selection criteria for the best model from each DL algorithm are based on the minimum training and validation losses and the highest accuracy of the validation dataset. After the training and validation process, the best model from

each DL algorithm is saved with its optimal hyperparameters. At the end, the selected models were evaluated on the test dataset using metrics such as precision, recall, F1-score, and area under the Receiver Operating Characteristic (ROC) curve (i.e., AUC).

### 3.5.1. Convolutional neural networks (CNN).

The CNN architecture utilized in this study extracted the temporal features from time-series IMU data using several convolutional layers [37]. Each convolutional layer is followed by batch normalization and ReLU activation. Although the general architecture remains consistent across both datasets, specific hyperparameters like the number of filters and fully connected nodes are fine-tuned to enhance performance for each dataset.

The initial step for the optimal models in both datasets involve adding a batch normalization layer to stabilize the input signal. Subsequently, there are several Conv1D layers with varying filter sizes and numbers: GSTRIDE utilizes nine Conv1D layers, while FRAILPOL utilizes three Conv1D layers. Batch normalization and ReLU activation are connected to each convolutional layer to ensure consistent feature extraction from the data, i.e., IMU signals. After the convolutional layers, the multidimensional output is converted to a 1D vector using the flattening layer. The resulting output is then fed into a dense layer, which is followed by a two-node output layer activated with the *'sofmax'* function. The CNN model architectures for both datasets are presented in the supplementary information as S1 Fig and S2 Fig. The optimal hyper-parameters of CNN models for both datasets are listed in Table 2.

### 3.5.2. Deep convolutional LSTM (DeepConvLSTM).

A DeepConvLSTM network [38] is a combination of convolutional and LSTM layers. It captures spatial and temporal information from the time-series data, i.e., IMU data. The best selected models were initialized with a batch normalization layer to stabilize the input signals, followed by a reshape layer to prepare the input for convolutional layers. Following that, the 2D convolutional layers with varying filter sizes (i.e., seven for GSTRIDE and two for FRAILPOL) were used in combination with batch normalization and ReLU activation. These layers ensured the extraction of robust spatial features.

The output obtained from the convolution layers in both datasets was transformed into a 2D tensor. Which is then fed into LSTM layers for the extraction of temporal features. In the GSTRIDE dataset, the LSTM network consists of a single LSTM layer with 55 units. Whereas, in the FRAILPOL dataset, five LSTM layers with various nodes were employed. These layers indicate the necessity of complexity in learning temporal features. After the LSTM layers, dropout is used to avoid the model's overfitting. The DeepConvLSTM architecture is concluded with a time-distributed dense layer that permits successive predictions, followed by a two-node output layer with the *'softmax'* activated function for a classification task. DeepConvLSTM architectures for both datasets provided in the supplementary information (S3 Fig and S4 Fig). The optimal hyperparameters of the best models are shown in Table 3.

### 3.5.3. InceptionTime.

The InceptionTime architecture [39] consists of a sequence of inception modules that extract the spatio-temporal features from both datasets. Inception modules perform multiple 1D convolutions with different kernel sizes in parallel with the same input.

The model configuration for GSTRIDE consists of six channels and six inception modules. It also features a 200-time step input layer with three Conv1D layers of varying kernel sizes, designed to capture different temporal characteristics.

**Table 2. Hyperparameters of CNN algorithm for both GSTRIDE & FRAILPOL datasets.**

| Hyperparameter | GSTRIDE | FRAILPOL |
|---|---|---|
| Input Dimensions | 200-time steps, 6 channels | 200-time steps, 7 channels |
| Learning Rate | 0.000332 | 0.001180 |
| Regularization Rate | 0.002428 | 0.000805 |
| Filters | [11, 65, 14, 17, 45, 58, 77, 50, 69] | [96, 52, 50] |
| FC Hidden Nodes | 638 | 224 |

**Table 3. Hyperparameters of DeepConvLSTM algorithm for both GSTRIDE & FRAILPOL datasets.**

| Hyperparameter | GSTRIDE | FRAILPOL |
|---|---|---|
| Input Dimensions | 200-time steps, 6 channels | 200-time steps, 7 channels |
| Learning Rate | 0.024103 | 0.037236 |
| Regularization Rate | 0.000107 | 0.000437 |
| Conv2D Filters | [92, 71, 47, 91, 81, 77, 64] | [42, 38] |
| LSTM Units | [55] | [21, 58, 72, 21, 88] |

On the other hand, the optimal setup for FRAILPOL shares a similar input layer but incorporates three inception modules with two Conv1D layers, seven channels, and a simpler design. Both models follow the same data processing steps, where the results of these convolutions are combined, batch normalized and passed through a ReLU activation. Lastly, global average pooling is carried out before a dense *'softmax'* layer is utilized for classification. Both models effectively extracted notable features from their respective datasets, ultimately leading to accurate classification results. The supplementary information contains the InceptionTime model architectures for both datasets: S5 Fig illustrates the architecture for FRAILPOL dataset, while S6 Fig shows the architecture for GSTRIDE. The optimal hyperparameters of the best models are listed in Table 4.

## 4. Results

Precision, recall, F1-score, and accuracy are the key evaluation metrics used to evaluate the performance of DL algorithms on the test dataset. Furthermore, the performance of DL models was validated using the area under the ROC curve (i.e., AUC). The confusion matrices provide a detailed model's performance across the classes. Training and validation losses were the key evaluation criteria to validate and select the best model from each DL algorithm.

The best-performing architecture for each model type (i.e., CNN, DeepConvLSTM, and InceptionTime), was evaluated three times independently. For each execution, the dataset was partitioned into training and validation sets using the proposed participant-based stratified split. Subsequently, the models were trained from scratch (using Mcfly library), and final performance was evaluated on the same held-out test set.

To start with the training results, the best models from each DL algorithm were selected based on the minimum training and validation losses with the highest accuracy. As shown in Fig 4, the InceptionTime algorithm reported the best performance results on both datasets, i.e., training and validation losses of 0.3802 and 0.4275, respectively, on the FRAILPOL dataset. Whereas on the GSTRIDE dataset, the training and validation losses were 0.1118 and 0.1307, respectively. The best performance of the InceptionTime algorithm on the training sets proved its ability to generalize and efficiently learn across diverse datasets.

**Table 4. Hyperparameters of InceptionTime algorithm for both GSTRIDE & FRAILPOL datasets.**

| Hyperparameter | GSTRIDE | FRAILPOL |
|---|---|---|
| Input Dimensions | 200-time steps, 6 channels | 200-time steps, 7 channels |
| Inception Modules | 6 | 3 |
| Conv1D Layers/Module | 3 | 2 |
| Kernel Sizes | 3, 5, 11 | 3, 5 |
| Learning Rate | 0.0918 | 0.0135 |
| Regularization Rate | 0.0135 | 0.0328 |
| Number of Filters | 69 | 85 |
| Max Kernel Size | 67 | 24 |

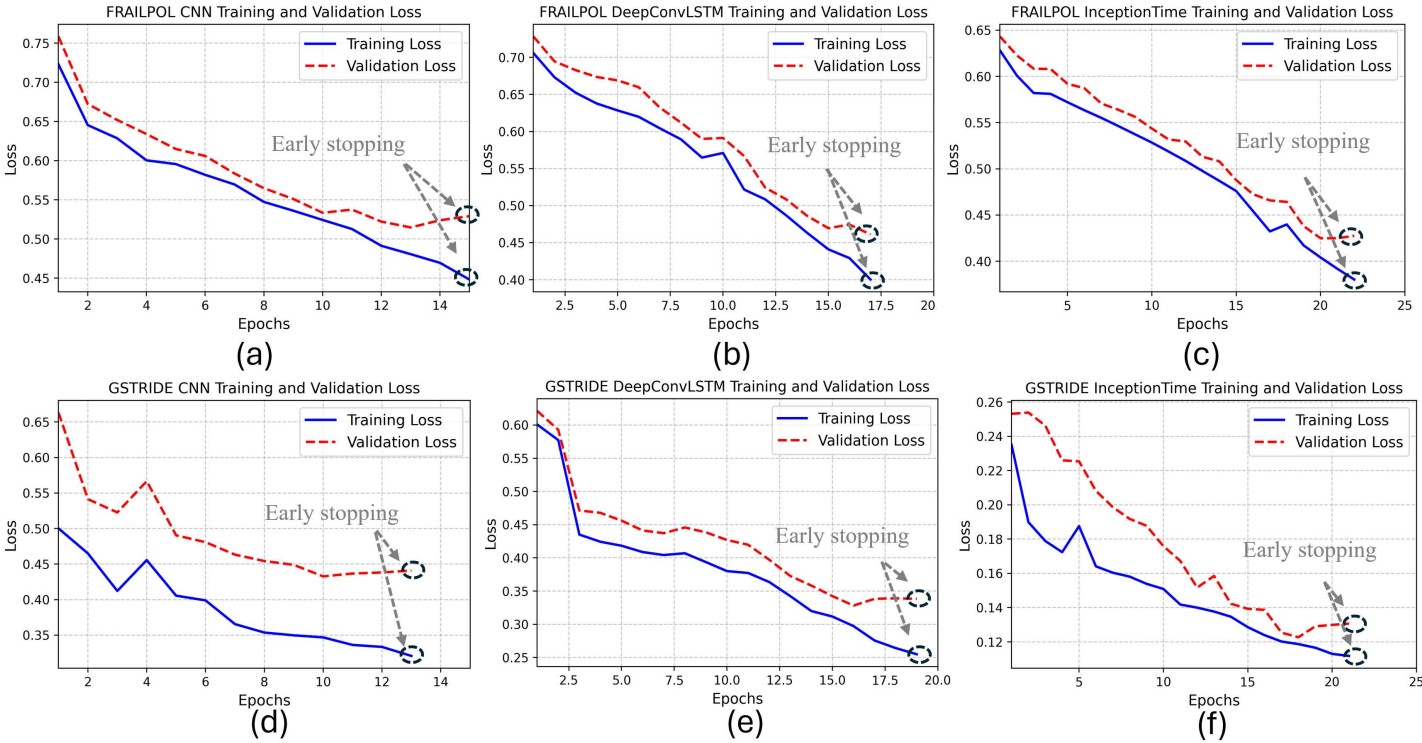

**Fig 4. Training and validation losses of DL algorithms on FRAILPOL dataset: (a) CNN, (b) DeepConvLSTM, (c) InceptionTime; and on GSTRIDE dataset: (d) CNN, (e) DeepConvLSTM, (f) InceptionTime.**

**Table 5. Overall classification results of DL algorithms on both GSTRIDE & FRAILPOL datasets across three independent runs (Mean ± Standard Deviation).**

| Model | Dataset | Precision | Recall | F1-Score | Accuracy |
|---|---|---|---|---|---|
| **CNN** | GSTRIDE | 0.78 ± 0.02 | 0.78 ± 0.01 | 0.78 ± 0.03 | 0.77 ± 0.02 |
| **DeepConvLSTM** | | 0.75 ± 0.01 | 0.74 ± 0.02 | 0.75 ± 0.01 | 0.76 ± 0.01 |
| **InceptionTime** | | **0.81 ± 0.01** | **0.83 ± 0.01** | **0.82 ± 0.01** | **0.82 ± 0.01** |
| **CNN** | FRAILPOL | 0.76 ± 0.01 | 0.72 ± 0.02 | 0.75 ± 0.02 | 0.74 ± 0.02 |
| **DeepConvLSTM** | | 0.74 ± 0.02 | 0.73 ± 0.01 | 0.75 ± 0.01 | 0.76 ± 0.01 |
| **InceptionTime** | | **0.76 ± 0.01** | **0.77 ± 0.01** | **0.78 ± 0.01** | **0.79 ± 0.01** |

InceptionTime algorithm also outperformed on both datasets in the testing phase. As in Table 5, the reported metrics represent the mean and standard deviation across three independent runs for each model on the GSTRIDE and FRAIL-POL test sets. InceptionTime achieves the highest mean accuracy and F1-score of 0.82 (standard deviation ± 0.01) for the GSTRIDE dataset, while it has mean accuracy of 0.79 (standard deviation ± 0.01) and a mean F1-score of 0.78 (standard deviation ± 0.01) for the FRAILPOL dataset. Following InceptionTime, DeepConvLSTM demonstrated a mean accuracy of 0.76 for both datasets and a mean F1-score of 0.75. CNN has shown a slightly lower performance with a mean accuracy and F1-score of 0.77 and 0.74, respectively, for the GSTRIDE dataset. Similarly, for the FRAILPOL dataset, the mean accuracy and F1-score of 0.78 and 0.75, respectively, were reported.

Accurate classification of frail and non-frail individuals is critical in clinical settings, especially given patient variability. The effectiveness of a participant-based data split demonstrates its practical significance as illustrated in the window-level

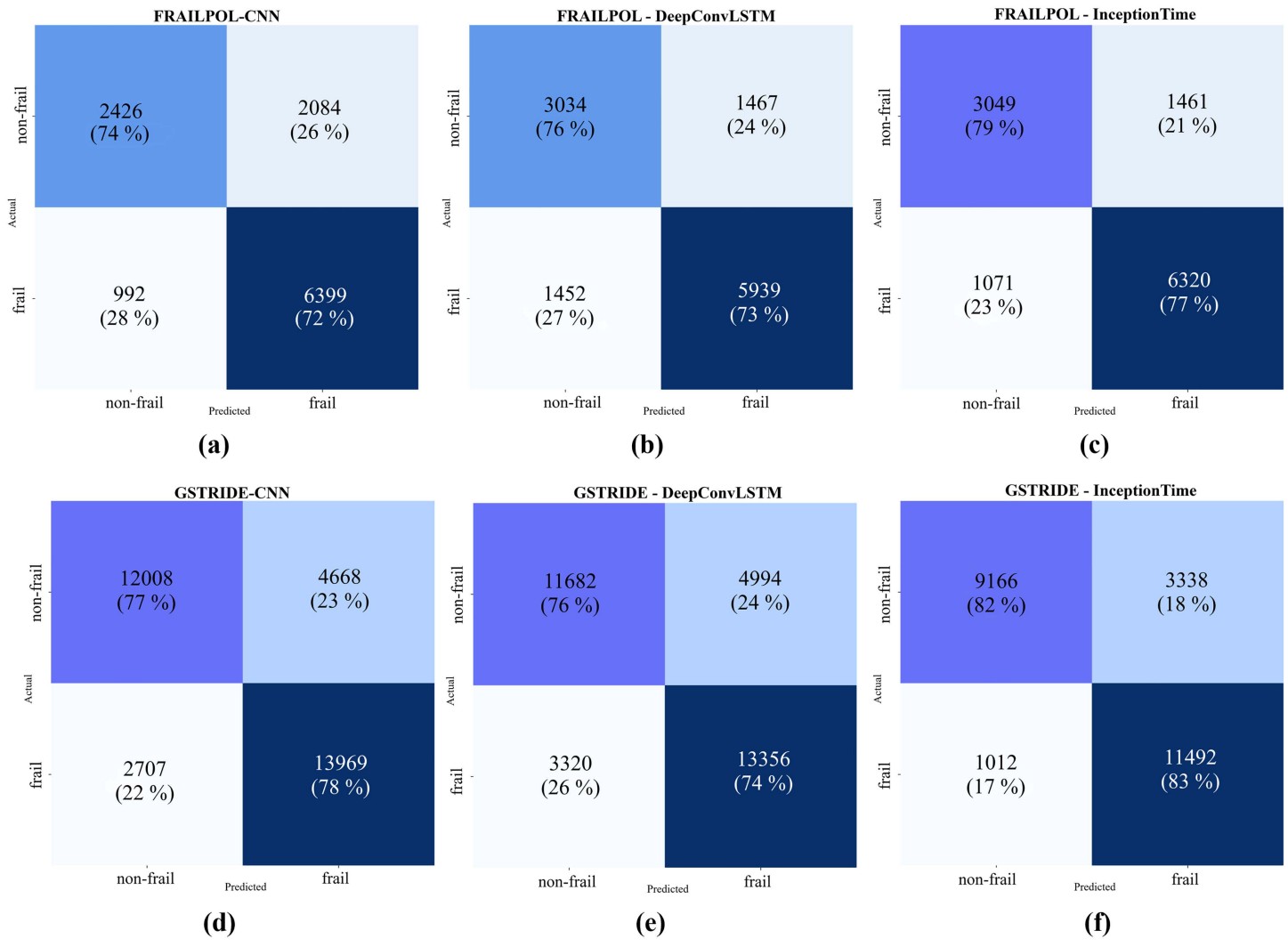

**Fig 5. Window-level confusion matrices for (a) (d) CNN (b) (e) DeepConvLSTM, and (c) (f) InceptionTime models evaluated on both datasets: FRAILPOL and GSTRIDE.**

confusion matrices (Fig 5). For comprehensive analysis, we also calculated confusion matrices at the subject-wise confusion matrices (as shown in supporting information (S7 Fig)) in addition to the window-level evaluation. This ensures that classification performance was evaluated using statistically independent metrics. The InceptionTime method correctly classified frail and non-frail stages in both datasets demonstrates its potential for accurate frailty classification, which is critical for successful clinical evaluation and intervention.

## 5. Discussion

Analysis of confusion matrices at both the window and subject levels (Fig 5 and S1 Fig) confirms that the proposed framework gives more consistent performance. The window-level confusion matrices (Fig 5) show that InceptionTime outperforms other models in both datasets. On the FRAILPOL dataset, InceptionTime correctly classified 77% of frail windows

compared to 72% and 73% for CNN and DeepConvLSTM, respectively. Whereas on the GSTRIDE dataset, InceptionTime achieved a significantly lower false negative rate for frail windows (17%) compared to the CNN (22%) and DeepConvLSTM (26%) baselines.

The subject-wise confusion matrices (S1 Fig) further validate the performance of the proposed framework. On the FRAILPOL dataset, InceptionTime correctly identifies 78% of frail participants compared to only 71% and 73% for the CNN and DeepConvLSTM, respectively. Furthermore, the false rate for the frail class is reduced on the GSTRIDE dataset (i.e., 20%) for InceptionTime. While CNN and DeepConvLSTM misclassified 2 frail participants out of 5. InceptionTime achieved the highest true positive rates for both frail and non-frail individuals and a balanced number of false positives and negatives, proving its ability to handle diverse raw IMU data.

The Receiver Operating Characteristic (ROC) curves for all DL models for the FRAILPOL and GSTRIDE datasets are shown in S8 Fig. For the FRAILPOL dataset, the InceptionTime model outperformed with an Area Under the Curve (AUC) ($\approx 0.75$–$0.80$). It indicates that the non-frail and frail classes are separated effectively by achieving a high true positive rate without a substantial increase in the false positive rate. The DeepConvLSTM demonstrates consistently higher true positive rates with an AUC $\approx 0.74$–$0.76$. However, the CNN shows a moderate AUC of $\approx 0.70$–$0.74$ with noticeable overlaps between the frail and non-frail classes. This overlap indicates limited separability, particularly at lower false positive rates.

Similarly, the InceptionTime model achieved the best performance on the GSTRIDE dataset with an AUC of $\approx 0.80$–$0.82$. Subsequently, the CNN model achieved an AUC of $\approx 0.76$–$0.79$, indicating a significant performance gain compared to its performance with the FRAILPOL dataset. While the DeepConvLSTM model showed stable performance with an AUC of $\approx 0.73$–$0.75$. The superior performance of DL models with the GSTRIDE dataset is due to its richer temporal representation per participant. There are more data points per participant, which improves class separability and allows the models to learn more stable gait patterns. Moreover, the balanced ROC curves for non-frail and frail classes indicate that the performance gains are not driven by bias toward a single class.

The CNN and DeepConvLSTM models show effective performance on classifying non-frail individuals, they show comparatively reduced sensitivity with frail individuals due to large false negative rates. This difference is possibly due to class imbalance. The class imbalance was addressed through stratified participant sampling and by evaluating models' performance using class-specific metrics (precision, recall, F1-score) rather than accuracy alone. Additionally, the findings indicate that the InceptionTime model performed more effectively on the GSTRIDE dataset than on FRAILPOL. This is mainly due to the greater number of data samples collected from longer gait distances, which emphasizes the importance of gait length in improving classification accuracy.

Many previous studies analyzed the frailty syndrome, particularly frailty classification [14–17] and fall risks [20–27] among different participants. However, direct comparison of results is challenging because the previous studies trained the DL models on raw IMU gait data by first applying the data segmentation techniques (i.e., sliding window) to segment the data, followed by different data splitting techniques such as random data splitting, K-fold or leave-one-out cross validation, etc. These approaches may cause a problem of data leakage, which leads to model overfitting and reduces generalizability due to the small dataset and lack of temporal separation between training and testing sets. The findings show that the previous studies achieved higher or comparable results but utilized limited and less diverse datasets. The utilization of random data partitioning techniques and smaller datasets is mainly the reason for these outcomes.

This study introduces a DL framework to overcome these issues. In the proposed framework, it makes sure that no data from the same participant is present in both training and testing sets, unlike the random data partitioning utilized in the studies [14–17,20–27], which can result in data leaking and overfitting. As in the study [14], the authors applied a random data partitioning approach to 20 participants, achieving 95% accuracy with an LSTM-CNN model for frailty classification. Another study [21] reported an F1-score of 95.18% with a CNN-LSTM model but utilized random data partitioning with cross-validation on a small dataset, but this study evaluated fall risk. Similarly, the studies [22,24] also evaluated fall risk by employing a random data partitioning approach. The study [22] implemented the CNN-BiLSTM model and

achieved an accuracy of 98.40%. Whereas in study [24], LSTM model with transfer learning approach was used and reported an F1-score of 93%. These models may have learned the participant-specific features if participant data is not properly divided between training and testing sets, which restricts the models' generalizability to a larger and more diverse dataset.

The proposed data partitioning framework that is integrated with DL algorithms is evaluated on two diverse datasets to ensure that the optimal model (i.e., InceptionTime) is robust and generalizable to different participants. The utilization of varied data partitioning methodologies and datasets makes the direct comparison with previous studies challenging. However, this study addresses the critical issue of data leakage and model overfitting in previous studies. It ensures that no participant's windows data overlaps between training and testing sets. The proposed approach provides a more reliable evaluation of the DL model's performance for frailty assessment in real-world clinical settings.

### 5.1. Limitations and future work

In future work, more diversified datasets from different populations can be considered for more accurate frailty analysis. Additionally, more sensors that can capture environmental data can also help in understanding frailty factors. More advanced signal processing techniques and DL algorithms should be explored to improve the accuracy of frailty detection systems. A real-time application with a user-friendly interface for clinical setup is required to implement the frailty assessment advancements into practical tools for efficient patient care.

## 6. Conclusion

A DL framework is proposed in this study that utilizes the combination of the sliding window technique and a participant-based data partitioning approach. The goal was to classify the frailty based on the raw IMU signals, which reflect a real-world clinical scenario. Three DL algorithms were evaluated on two different datasets. The datasets are diversified by the sensor count and the mounting position of the IMU sensors on participant body parts. The InceptionTime algorithm outperformed on both datasets, which highlights its capacity to capture intrinsic spatio-temporal features from single and multiple sensor data. The results also prove the InceptionTime algorithm's effectiveness in classifying the signal segments as frail or non-frail (robust). This enables an objective frailty assessment method for early detection and intervention of frailty in older adults.

This study also highlights the critical issue of data leakage when using a data segmentation technique with a random data partitioning approach for raw IMU signal data. Previously, the studies [14–17,20–27] used the random data partitioning technique, which causes a problem of overlapping data between the training and testing sets. This leads to data leakage and model overfitting. The proposed approach solved this issue. However, its clinical translation requires rigorous validation across more diverse demographic and clinical populations. Furthermore, integrating multimodal data streams is essential for developing robust and generalizable systems capable of operating effectively in real-world environments.

### Supporting information

**S1 Fig. Architecture of the CNN model for frailty classification on the FRAILPOL dataset.**
(TIFF)

**S2 Fig. Architecture of the CNN model for frailty classification on the GSTRIDE dataset.**
(TIFF)

**S3 Fig. Architecture of the DeepConvLSTM model for frailty classification on the FRAILPOL dataset.**
(TIFF)

**S4 Fig. Architecture of the DeepConvLSTM model for frailty classification on the GSTRIDE dataset.**
(TIFF)

**S5 Fig. Architecture of the InceptionTime model for frailty classification on the FRAILPOL dataset.**
(TIFF)

**S6 Fig. Architecture of the InceptionTime model for frailty classification on the GSTRIDE dataset.**
(TIFF)

**S7 Fig. Subject-wise confusion matrices for (a)(d) CNN (b)(e) DeepConvLSTM, and (c)(f) InceptionTime models evaluated on both datasets: FRAILPOL and GSTRIDE.**
(TIFF)

**S8 Fig. Receiver Operating Characteristic (ROC) curves for DL models: (a)(d) CNN, (b)(e) DeepConvLSTM, and (c)(f) InceptionTime evaluated on both FRAILPOL and GSTRIDE datasets, illustrating the discriminative performance across frail and non-frail classes.**
(TIFF)

## Acknowledgments

This work was supported by the Department of Computer Graphics, Vision, and Digital Systems, under the statutory research project for young scientists (Rau6, 2025), Silesian University of Technology, Gliwice, Poland.

## Author contributions

**Conceptualization:** Arslan Amjad.

**Data curation:** Jerzy Sacha, Magdalena Sacha, Piotr Feusette, Wojciech Wolański, Mariusz Konieczny, Zbigniew Borysiuk.

**Formal analysis:** Agnieszka Szczęsna, Monika Błaszczyszyn.

**Methodology:** Arslan Amjad, Agnieszka Szczęsna.

**Project administration:** Agnieszka Szczęsna.

**Resources:** Basheir Khan.

**Software:** Arslan Amjad, Basheir Khan.

**Supervision:** Agnieszka Szczęsna, Monika Błaszczyszyn.

**Validation:** Agnieszka Szczęsna, Monika Błaszczyszyn.

**Visualization:** Basheir Khan.

**Writing – original draft:** Arslan Amjad.

**Writing – review & editing:** Agnieszka Szczęsna.

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
