## [Decision Letter · Decision Letter 0]

4 Nov 2025

Dear Dr. Amjad,

Thank you for submitting your manuscript to PLOS ONE. After careful consideration, we feel that it has merit but does not fully meet PLOS ONE’s publication criteria as it currently stands. Therefore, we invite you to submit a revised version of the manuscript that addresses the points raised during the review process.

We look forward to receiving your revised manuscript.

Kind regards,

Anne E. Martin

Academic Editor

PLOS ONE

Journal Requirements:

3. In the online submission form, you indicated that “The data supporting the findings of this study are from two sources. The GSTRIDE dataset is publicly available at https://zenodo.org/records/6883292 (García-de-Villa, S., et al. (2023)). The second dataset, FRAILPOL, is proprietary dataset generated for this study. The dataset is not publicly is not publicly available but can be obtained from the corresponding author upon reasonable request for research purposes.”

Additional Editor Comments:

Please address all reviewer comments, paying particular attention to Reviewer 1's concerns.

Reviewer's Responses to Questions

**Comments to the Author**

1. Is the manuscript technically sound, and do the data support the conclusions?

Reviewer #1: Partly

Reviewer #2: Yes

Reviewer #3: Partly

2. Has the statistical analysis been performed appropriately and rigorously?

Reviewer #1: No

Reviewer #2: N/A

Reviewer #3: I Don't Know

3. Have the authors made all data underlying the findings in their manuscript fully available?

Reviewer #1: No

Reviewer #2: Yes

Reviewer #3: No

4. Is the manuscript presented in an intelligible fashion and written in standard English?

Reviewer #1: Yes

Reviewer #2: No

Reviewer #3: Yes

Reviewer #1: The manuscript explores a relevant topic: frailty classification using deep learning and IMU data. The study is conceptually sound and addresses an important problem in clinical gait analysis. However, several important deficiencies prevent the work from meeting technical and reproducibility standards.

- Data availability:

The study states that the data are ‘fully available,’ but clarifies that the FRAILPOL dataset can only be obtained ‘upon reasonable request.’ This does not guarantee the repeatability of the results.

I recommend making the FRAILPOL dataset public, at least in anonymized form, in a recognized repository (e.g. Zenodo, Figshare, OSF or PhysioNet), including the processed IMU files and participant metadata (age, sex, FI, sensor type, etc.).

If there are ethical or legal restrictions, the authors should clearly justify these limitations and provide a transparent mechanism for access.

Section 3. Methodology

The description of the methodological framework is too general and should be developed more precisely to ensure the transparency and reproducibility of the study. We recommend:

Section 3.3 Data Pre-Processing

- Specify what type of filter or noise reduction method was applied to the IMU signals during preprocessing, noting the main parameters (order, cutoff frequency, etc.).

- Explain more clearly the order in which time segmentation using the sliding window technique and participant-centric data partitioning are applied, indicating precisely how both stages are integrated into the workflow.

- Provide the specific time parameters for the windows (e.g., how many seconds a 200-sample window represents and the justification for the stride of 50), also explaining whether other configurations were evaluated.

Section 3.4 Data Partitioning

The manuscript indicates that 10 of the 163 participants in the GSTRIDE set (and 100 of the 682 in FRAILPOL) were selected to form the test set. However, it does not explain how these subjects were chosen or whether the procedure was repeated multiple times. If the selection were performed only once, the results could depend on that particular partition, compromising the statistical robustness and external validity of the model. It is recommended to describe in detail the selection criterion (random or stratified by class) and, preferably, apply per-participant cross-validation (e.g., k-fold) to obtain more stable and representative performance metrics. Reporting the mean and standard deviation of the metrics across different partitions would significantly strengthen the credibility of the results.

- Detail how the participants who make up the training, validation, and test sets were selected, indicating whether the assignment was random or stratified (by age, gender, or frailty level) and whether a balance between classes (frail/non-frail) was maintained in each set.

- Specify whether cross-validation or repetition of the process was applied to assess the stability and robustness of the results.

Section 3.5 Deep Learning Methods and Implementation

Although the three architectures used (CNN, DeepConvLSTM, and InceptionTime) are adequately presented, the technical description is primarily descriptive and lacks justification for the design and optimization decisions. It would be advisable to explain why these architectures were selected, how the hyperparameters were tuned (whether through systematic or automatic search), and whether training sessions were repeated with different seeds to ensure stable results.

Section 4. Results

The results section adequately presents the basic performance metrics (precision, recall, F1 score, and accuracy) along with loss curves and confusion matrices. However, there are aspects that need to be improved:

- The evaluation metrics are reported as single values without confidence intervals or standard deviations, which makes it difficult to estimate performance variability.

- Figure 5, using a three-dimensional graph, is difficult to read and does not reflect the variability between models; it would be preferable to replace it with a 2D bar chart or boxplots with error bars that allow a quantitative comparison of the results.

- Confusion matrices (Figure 6) are useful for visualizing classification balance between classes, but they present a significant conceptual problem:

o The matrices are presented as absolute values, which makes it difficult to compare datasets of different sizes and models. It would be advisable to normalize the matrices by rows or columns (as a percentage).

o The counts are on the order of tens of thousands of instances, even though the test set consists of only 10 participants in GSTRIDE and 100 in FRAILPOL. This indicates that the metrics were calculated at the time window level rather than at the participant level, implying that multiple non-independent observations from the same individual were treated as separate samples. This approach can artificially inflate accuracy and other metrics, compromising the statistical validity of the results. It is recommended to recalculate the metrics considering the actual experimental unit (the participant), or at least report the results from both perspectives (by window and by subject). Metrics should be calculated per subject (averaging across windows) and then averaged across subjects, not directly over all windows together.

Section 5. Discussion

The discussion interprets the results descriptively, but without quantitative analyses to support the conclusions. The improved performance of InceptionTime is attributed to gait length and the proposed partitioning framework, without formally proving it. Furthermore, the metrics are computed at the window level, which limits claims about generalization. The limitations section is too general and should address the most relevant methodological issues.

Section 6. Conclusion

The conclusion summarizes the study's objectives well, but the claims about the model's efficacy and generalizability are too strong considering the available evidence. It is recommended that these statements be expressed with greater caution and that the need for further validation at the participant level and in broader populations be emphasized.

Reviewer #2: This study proposed to use a participant-based data partitioning method to prevent data leakage, improving the generalizability of DL models for clinical applications. Various architectures (CNN, DeepConvLSTM, and InceptionTime) were used and experiments on two datasets (GSTRIDE and FRAILPOL) demonstrate the effectiveness of the proposed method. However, several questions are still needed to be addressed:

1. Current versions include numerous instances of awkward phrasing, grammatical errors, and unclear sentences. Some examples are provided here: “The CNN architecture [36] utilized in this study extracted the temporal features…”; Use of “sofmax” in CNN output layer description; “frailty classification from Gait Signals” → “gait signals” should be lowercase. A comprehensive language revision is strongly recommended.

2. While two datasets were used in this study, one of them is not publicly available. The authors are encouraged to use more public datasets to validate the generability of the proposed data partitioning method. In addition, a summary table or schematic diagram for each model architecture used in this study will be beneficial for model architecture clarification.

3. The study did not explore which features or sensor placements contributed most to the classification, further ablation studies are strongly recommended to include.

4. Please address or discuss the below limitations in the manuscript:

[1] Limited Diversity in Demographics and Sensor Placement from datasets.

[2] Consider elaborating on how real-time implementation might be approached, given the focus of this manuscript is on clinical applicability. Ideally, computational complexity should be included and evaluated in this study.

[3] The manuscript does not clearly discuss how the proposed method can address the imbalanced dataset issue that might affect model learning performance.

Reviewer #3: In this paper, Amjad et al. proposed an advanced frailty assessment method combining wearable sensors with Deep Learning (DL) techniques to classify individuals into frail or non-frail stages. Two diverse datasets, i.e., GSTRIDE and FRAILPOL, were utilized for enhanced frailty analysis, employing one Inertial Measurement Unit (IMU) sensor and five IMU sensors with varying configurations and mounting positions. The proposed DL framework utilizes the combination of the sliding window technique and a participant-based data partitioning approach. The goal was to classify the frailty based on the raw IMU signals. Three DL algorithms—CNN, DeepConvLSTM and InceptionTime, were evaluated. The authors found that InceptionTime works the best.

I have a few comments:

Suggest removing Fig 4, training errors are not very informative for understanding the performance of ML algorithms.

Table 4. I am surprised to see there is very little difference across the four types of measures for all datasets and methods compared. Please explain why this is the case. Also, because the sample size of FRAILPOL is much bigger than that of GSTRIDE, I am surprised to see the performance is actually worse. Please explain why this is the case.

Fig 5. Why only compare accuracy? How about other measures?

Fig 6. It is fine to use confusion matrix, but a threshold is needed to construct such a matrix. But no information is provided on the threshold and why choose that threshold.

I suggest authors to draw ROC curves to compare performance visually.

**Do you want your identity to be public for this peer review?** For information about this choice, including consent withdrawal, please see our Privacy Policy

Reviewer #1: No

Reviewer #2: No

Reviewer #3: No

---

## [Author Response · Author response to Decision Letter 1]

30 Dec 2025

Dear Editor,

Thank you for allowing the resubmission of our manuscript, with an opportunity to address the reviewers’ comments. We appreciate the constructive feedback and have carefully revised the manuscript to address all points raised. The reviewers’ suggestions have substantially strengthened the methodology, analysis, and clarity of our work.

Our detailed point-by-point response to each comment is attached to a file (Response to Reviewers), outlining the specific changes implemented in the revised manuscript. All major revisions have been highlighted in the revised manuscript file for your convenience.

We believe the manuscript has been significantly improved and hope it now meets the journal’s standards for publication.

---

## [Decision Letter · Decision Letter 1]

13 Jan 2026

Dear Dr. Amjad,

Thank you for submitting your manuscript to PLOS ONE. After careful consideration, we feel that it has merit but does not fully meet PLOS ONE’s publication criteria as it currently stands. Therefore, we invite you to submit a revised version of the manuscript that addresses the points raised during the review process.

plosone@plos.org . A letter that responds to each point raised by the academic editor and reviewer(s). You should upload this letter as a separate file labeled 'Response to Reviewers'.A marked-up copy of your manuscript that highlights changes made to the original version. You should upload this as a separate file labeled 'Revised Manuscript with Track Changes'.An unmarked version of your revised paper without tracked changes. You should upload this as a separate file labeled 'Manuscript'.

We look forward to receiving your revised manuscript.

Kind regards,

Alessandro Mengarelli

Academic Editor

PLOS One

**Journal Requirements:**

**Additional Editor Comments:**

The reviewers were not fully satisfied by the replies and modifications of the paper. They still raised concerns about some points of the paper that deserve to be carefully taken in charge by the authors, since they involve crucial methodological points. I would suggest reply to every single reviewer's comment, producing the suggested additional analyses/results or providing well grounded rebuttal if the required changes/additional analyses cannot be done or are not appropriate in your opinion.

Reviewers' comments:

Reviewer's Responses to Questions

**Comments to the Author**

Reviewer #1: All comments have been addressed

Reviewer #2: All comments have been addressed

Reviewer #3: (No Response)

2. Is the manuscript technically sound, and do the data support the conclusions?

Reviewer #1: Partly

Reviewer #2: Yes

Reviewer #3: No

3. Has the statistical analysis been performed appropriately and rigorously?

Reviewer #1: No

Reviewer #2: N/A

Reviewer #3: No

4. Have the authors made all data underlying the findings in their manuscript fully available?

Reviewer #1: Yes

Reviewer #2: Yes

Reviewer #3: No

5. Is the manuscript presented in an intelligible fashion and written in standard English?

Reviewer #1: Yes

Reviewer #2: Yes

Reviewer #3: No

Reviewer #1: The authors have addressed the issues raised in the first review. The introduction of a participant-based data partitioning framework represents a clear methodological improvement, and the rationale for independent subject division is now well motivated and discussed consistently throughout the manuscript.

The methodology is clearer, including a more detailed description of pre-processing, model training and evaluation, as well as repeated experimental runs. The inclusion of the new confusion matrices is positive and demonstrates an awareness of the limitations of window-level evaluation.

However, several issues remain to be resolved:

- First, the data partitioning procedure lacks complete transparency and consistency, particularly with regard to the exact number of participants used in the divisions and how stratification was performed at the subject level. This affects reproducibility and needs to be clarified.

- Second, although subject-level evaluation is mentioned, the main results and conclusions are still based primarily on window-level metrics. Given that the clinical objective is the classification of frailty at the individual level, more emphasis should be placed on subject-level performance and, possibly, treated as the primary endpoint.

- Thirdly, the use of the term “personalised” remains problematic, as the proposed framework does not include subject-specific modelling, adaptation or personalisation strategies. This constitutes a conceptual exaggeration that should be reconsidered in the title and throughout the manuscript.

Overall, the manuscript has improved significantly compared to the previous version, but the points mentioned above must be addressed to ensure methodological rigour, conceptual clarity, and alignment between the established clinical objectives and the evaluation strategy.

Reviewer #2: (No Response)

Reviewer #3: The authors essentially refuse to make any changes I suggested. This is unacceptable. The two classes are very balanced. instead of confusion matrix, I insist that ROC curves to be shown and compared.

**Do you want your identity to be public for this peer review?** For information about this choice, including consent withdrawal, please see our Privacy Policy

Reviewer #1: No

Reviewer #2: No

Reviewer #3: No

---

## [Author Response · Author response to Decision Letter 2]

19 Jan 2026

Thank you for allowing the resubmission of our manuscript, with an opportunity to address the reviewers’ comments. We appreciate the constructive feedback and have carefully revised the manuscript to address all points raised. The reviewers’ suggestions have substantially strengthened the methodology, analysis, and clarity of our work.

A response to the reviewers' file is attached, which addresses a point-by-point response to each comment, outlining the specific changes implemented in the revised manuscript. All major revisions have been highlighted in the revised manuscript file for your convenience.

---

## [Decision Letter · Decision Letter 2]

7 Feb 2026

A Deep Learning Framework for Gait-Based Frailty Classification Using Inertial Measurement Units

PONE-D-25-50242R2

Dear Dr. Amjad,

We’re pleased to inform you that your manuscript has been judged scientifically suitable for publication and will be formally accepted for publication once it meets all outstanding technical requirements.

Kind regards,

Alessandro Mengarelli

Academic Editor

PLOS One

Additional Editor Comments (optional):

Reviewers' comments:

Reviewer's Responses to Questions

**Comments to the Author**

Reviewer #2: All comments have been addressed

Reviewer #3: All comments have been addressed

2. Is the manuscript technically sound, and do the data support the conclusions?

Reviewer #2: (No Response)

Reviewer #3: Partly

3. Has the statistical analysis been performed appropriately and rigorously?

Reviewer #2: (No Response)

Reviewer #3: Yes

4. Have the authors made all data underlying the findings in their manuscript fully available?

Reviewer #2: (No Response)

Reviewer #3: Yes

5. Is the manuscript presented in an intelligible fashion and written in standard English?

Reviewer #2: (No Response)

Reviewer #3: Yes

Reviewer #2: (No Response)

Reviewer #3: (No Response)

**Do you want your identity to be public for this peer review?** For information about this choice, including consent withdrawal, please see our Privacy Policy

Reviewer #2: No

Reviewer #3: No

---

## [Editor Report · Acceptance letter]

PONE-D-25-50242R2

PLOS One

Dear Dr. Amjad,

I'm pleased to inform you that your manuscript has been deemed suitable for publication in PLOS One. Congratulations! Your manuscript is now being handed over to our production team.

Kind regards,

on behalf of

Dr. Alessandro Mengarelli

Academic Editor

PLOS One